# Influence of Optimum Particle Packing on the Macro and Micro Properties of Sustainable Concrete

**Wisam J. Abushama** [1], **Adil K. Tamimi** [1,*], **Sami W. Tabsh** [1], **Magdi M. El-Emam** [1], **Ahmad Ibrahim** [2] **and Taghreed Kh Mohammed Ali** [3]

1 Department of Civil Engineering, College of Engineering, The American University of Sharjah, Sharjah P.O. Box 26666, United Arab Emirates; b00020441@aus.edu (W.J.A.); stabsh@aus.edu (S.W.T.); melemam@aus.edu (M.M.E.-E.)

2 Elkem Materials, Dubai P.O. Box 262213, United Arab Emirates; ahmad.ibrahim@elkem.com

3 Department of Architecture Engineering, Faculty of Engineering, Koya University, Koya KOY45, Kurdistan Region, Iraq; taghreed.khaleefa@koyauniversity.org

* Correspondence: atamimi@aus.edu

**Abstract:** In this research, the possibility of making eco-friendly concrete from available materials in the local United Arab Emirates (UAE) market was investigated. Supplementary cementitious materials, such as ground granulated blast-furnace slag (GGBS) and silica fume (SF), were utilized for decreasing the cement quantity, enhancing the particle size distribution and improving packing. In sum, 130 concrete specimens—cubes, cylinders, and prisms—from 10 different concrete mixes were tested to determine the enhancement levels in the fresh and hard properties of new concrete. The results showed the improved particle packing of the concrete, especially within the region of sizes 100–10,000 microns, produced by the Elkem Materials Mix Analyser (EMMA), closely matching the Andreassen theoretical model. The green concrete incorporating SF and GGBS possessed air content in the range 1.0–1.4% and compressive strength that is on average 11% higher than the well-packed concrete that did not contain SF or GGBS. Compared to the ACI 318 code's predictions, the experimental findings of the optimally packed concrete's moduli of rupture and elasticity were underestimated by 55–69% and 0.8–8.8%, respectively. The rapid chloride permeability test (RCPT) showed results as low as 392 coulombs for mixes with supplementary cementitious materials, indicating very low chloride permeability. Microstructural analysis using a scanning electron microscope (SEM) demonstrated that concrete with supplementary materials has fewer voids, more homogeneous integration of ingredients, and an abundance of C-S-H products that supported the RCPT findings and tests of mechanical properties. The study demonstrated a significant decrease in carbon dioxide ($CO_2$) emissions of concrete utilizing GGBS and SF and the financial feasibility of eco-friendly concrete in the UAE.

**Keywords:** eco-friendly concrete; particle packing; rapid chloride permeability; microstructural analysis; $CO_2$ emissions

## 1. Introduction

Concrete is considered the most utilized building construction material in the world. The manufacture of 1 ton of cement releases between 0.5 and 1 ton of carbon dioxide and consumes natural resources, such as limestone and clay. Additionally, a structure made of ordinary concrete (OC) can be permeable, which accelerates the corrosion of the steel rebars once subjected to severe climate exposure and leads to a reduction in the structure's durability and service life. Applying eco-friendly technologies in concrete design and production can resolve some of these problems and increase the structure's life span.

Over the years, published research has proven the benefits of using supplementary cementitious materials (SCMs) in concrete mixes, such as fly ash (FA), ground granulated blast-furnace slag (GGBS), and silica fume (SF). FA is a by-product of power plants that

generate electricity by burning pulverized coal, GGBS is a consequence of blast-furnaces that are used to make iron, and SF is an offshoot of producing silicon metal or ferrosilicon alloys. Multiple physical and mechanical properties of the resulting concrete can be improved with the use of such SCMs, leading to an eco-friendly product that results in a decrease in the use of natural materials. SCMs are added to concrete mixes for the sake of improving durability, reducing permeability, helping in pumpability, enhancing finishability in a plastic state, mitigating alkali reactivity, and improving the hardened mechanical properties of concrete through hydraulic or pozzolanic activity or both. However, the substitution of a portion of cement in concrete with SCMs may lead to delayed setting and low early strength. They are added to concrete in addition to or as a partial replacement of Portland cement and are usually considered part of the total cementing system.

Published research on particle packing has shown that by optimizing cement and aggregate particle size distribution, the voids between the particles can be significantly minimized, resulting in higher packing density and less binder required for filling pores. Efficient particle size distribution leads to improvement in the material strength, impermeability, and volume stability of the resulting product. Unlike the conventional method (CM) of concrete mix design, in the particle packing method (PPM), the smaller particles are selected to fill up the voids between large particles in order to increase the packing density.

## 2. Previous Studies

Contrary to previous research outcomes, the early effort by Jones et al. [1] showed that ternary blend concrete has higher rate of carbonation compared to normal concrete or concrete with FA. Later on, an experimental study by Babu and Kumar [2] indicated that there is no change in the compressive strength compared to the OC, and the maximum compressive strength was obtained at a replacement level between 25% and 50%. This was followed by the work of Khana and Lynsdaleb [3], who found that the addition of SF caused an increase in the depth of carbonation and improved the early strength of concrete, with the highest strength and lowest permeability values observed at SF content of 8–12% of cement. Menéndez et al. [4] demonstrated that an ordinary blend has the best compressive strength at 28 days, and blends with 20% and 35% GGBS possessed enhancement in strength compared with the normal one at 90 days. Dottoa et al. [5] found that SF can be efficiently utilized to protect reinforcement against corrosion, and the protection rate was directly proportional to the SF's content. Additionally, SF addition increased the compressive strength noticeably when compared to OC. Yang and Chiang [6] evaluated the relationship between the pore structure of concrete and the electrical charge that passes through concrete using RCPT and concluded that the continuity and the volume of pores increase with the rise in w/c ratio, and the relationship between compressive strength and pores quantity was linear. Results by Cheng et al. [7] on the influence of supplementary cementitious materials on durability of concrete showed that GGBS addition decreased the electrical current that passed through the specimens due to reduction in permeability, causing an increase in the flexural rigidity and deceleration in corrosion rate.

Oner and Akyuz [8] found that although the early strength of concrete incorporating GGBS was lower than that of OC, the strength in the long term increased considerably compared to normal mixes, with the best replacement of cement by GGBS for highest strength of 55 to 59. Nochaiya et al. [9] determined that adding of SF requires extra water to get the same slump, the setting time with SF was shorter than OC, the compressive strength of concrete containing SF and FA was enhanced by around 145%, and addition of FA and SF can cause denser microstructure. Khana and Siddique [10] noticed that adding SF decreased concrete permeability by up to more than 75%, upgraded the corrosion resistance of the reinforcement, and improved the transition phase of CH crystals by reducing the degree of orientation that enhances the mechanical properties of concrete and develops the formation of new C-S-H particles. Bagheri et al. [11] showed that while the replacement of cement with 15% GGBS led to a minor decrease in the compressive strength of concrete at different ages, adding SF was able to compensate for the strength of 15% slag mix to match the strength of

OC at 28 days and to exceed it at later ages. Ochbelagh et al. [12] suggested that adding 15% silica fume as a cement replacement can improve the microstructure of concrete and lead to increase in strength by around 22%. O'Connell et al. [13] investigated the performance of concrete containing GGBS against sulfates and acids available in waste treatment plants and found that adding GGBS to concrete led to a reduction in the expansion of concrete exposed to sulfate attack. The modulus of elasticity and strength of concrete having SF were studied by Saridemir [14], who confirmed that SF can improve the modulus of elasticity and found that there is a linear relationship between the compressive strength and modulus of elasticity of concrete.

Elchalakani et al. [15] showed that the compressive strength of concrete containing GGBS was close to that of ordinary concrete, although the setting time increased by 10–29%, the cost of producing eco-friendly concrete in the UAE was very close to the OC, and the best price was for concrete containing 80% GGBS. Abu Saleh [16] found that employing 70% GGBS in the mix can yield the highest compressive concrete strength, mixes with SF had higher compressive strength than those containing GGBS, and very low permeability results can be obtained from mixes having 70–100% GGBS and 10% SF. Tawfeeq et al. [17] found that the compressive and flexural concrete strengths were reduced by using GGBS and recycled block aggregates (RBA), and the minimum decrease was for mixes containing 25% GGBS, 25% RBA and 0.5% stapler's fibers. The flexural performance of reinforced concrete beams was improved by 27.7% for the mix containing 25% GGBS, 25% RBA and 1.5% steel fibers.

With respect to recent research on particle packing, Moinia et al. [18] utilized algorithms from previous research and found that concrete packing can enhance compressive strength. Pradhan et al. [19] observed that for the same aggregate type, there was significantly lower environmental impact for concrete proportioned with the particle packing mix (PPM) design method due to the need for less cement, and the lowest transport distance limit was observed for fossil fuels irrespective of the transportation scenario. Campos et al. [20,21] used PPM to produce low-cement high-strength concrete, found that the highest compressive strength values were obtained with 6% excess paste, and observed a tendency for reduced ultrasonic pulse propagation velocity and high electrical resistivity with 33% or more paste. Vatannia et al. [22] developed a cost-effective PPM procedure for making ultrahigh-performance (UHP) fiber-reinforced concrete, which confirmed that an increase in cement content can result in a reduction in strength when the combined particle size distribution contains too many particles of a similar size. Dingqiang et al. [23] utilized a D-optimal mixture design approach and genetic algorithm to develop superior mechanical properties of UHP concrete with relatively low cement content and silica fume due to its higher wet packing density and lower porosity. Alqamish and Tamimi [24] showed that the addition of nanosilica improved the microstructure and the interface structure of sustainable concrete due to its high pozzolanic activity and the nanofiller effect. Chu et al. [25] showed that proper proportioning of recycled aggregate using PPM in the concrete mix can increase the compressive strength of recycled aggregate concrete by 156%. Liu [26] researched the validity of Dewar's particle packing theory for recycled concrete aggregate and suggested that while the amount of powder has a limited influence on the packing density, the fine particles in the aggregate could have an important influence on the void ratio in the concrete mix. Londero [27] investigated a low-cement concrete mix by examining cyclic interactions and found that it was possible to reduce 40% of the cement consumption by applying particle packing techniques, leading to 28-day compressive and tensile strengths of 31 MPa and 2.9 MPa, respectively. With the help of the hot-pressing method and by optimizing insulating particle packing, Wei et al. [28] showed that the lightweight cementitious composites from glass, fly ash, cement, and silica fume have excellent mechanical properties and outstanding thermal insulation, with reasonable flexural strength, compressive strength, and thermal conductivity. García-Cortés et al. [29] showed a range of combinations in which the maximum packing density can be obtained, the theoretical models were overly complicated for easy application in comparison with

the experimental approach, and maximizing the amount of the larger aggregate fraction was beneficial for the concrete mix. Jiang et al. [30] demonstrated that the slump of fresh RAC increases in a growing rate with the cement paste volume, incorporation of optimum sand-to-aggregate volume ratio improves the packing density of total aggregate, and an excessive cement paste volume or sand-to-aggregate volume ratio negatively affects the microstructures of RAC, resulting in degradation in their mechanical properties. Laboratory experiments by Yin et al. [31] showed that the optimal volumetric steel fiber content of 2% can improve the packing density, raise the compressive strength of the concrete, and improve the microstructures. Research by Lv et al. [32] on the influence of fine aggregate particle packing on the paste threshold of self-compacting concrete indicated that when the maximum packing density of the fine aggregates increases, the paste can reach high flow performance with less yield stress and viscosity, leading to lower rheological thresholds. More recently, Sun et al. [33] developed a nonlinear packing model for porosity that can forecast the packing density of microaggregates and determined the correlation coefficients of the model based on granular soil, whereby the optimal parameter combinations were obtained through genetic algorithm inversion.

## 3. Objective

In the present study, a new modeling technique using Elkem Materials Mix Analyser (EMMA, Version 1) software [34] was utilized to improve the particle packing of concrete components that are expected to lead to enhancement in the rheology, mechanical properties, and durability aspects of eco-friendly concrete containing GGBS and SF. EMMA is an easy-to-use computer program that can assist professionals in optimizing their concrete mix proportions and designing self-flowing refractory castable compositions. In this approach, the user enters the particle size distribution (PSD) of the raw and additive materials and EMMA predicts the most favorable blend of those materials for producing concrete of target characteristics. Furthermore, EMMA can calculate the impact of producing a given concrete mixture on carbon dioxide ($CO_2$) emissions. To show these enhancements, microstructural analysis was performed with the help of a scanning electron microscope. Additionally, a cost analysis was carried out to explore the economic validity of using such eco-friendly concrete in the United Arab Emirates (UAE).

## 4. Investigation Plan

### 4.1. Characteristics of Materials

The constituents that were considered in the making of concrete in the study included OP cement, dune sand, crushed rock sand, 10 mm graded aggregate, 20 mm graded aggregate, GGBS and SF. All the materials were readily available locally in the UAE market. The specific gravity (SG) and chemical compositions of the used materials are presented in Tables 1 and 2, respectively, while the distribution analysis of GGBS particle size, fine aggregate, coarse aggregate, silica fume and cement is shown in Figure 1.

**Table 1.** Specific gravity of concrete ingredient.

| Material | Cement | Dune Sand | Crushed Rock Sand (5 mm) | Aggregates (10 mm) | Aggregates (20 mm) | GGBS | Silica Fume |
|---|---|---|---|---|---|---|---|
| Specific Gravity | 3.14 | 2.66 | 2.67 | 2.92 | 2.94 | 2.18 | 2.20 |

**Table 2.** Chemical composition of concrete ingredients.

| Chemical Composition | Cement (%) | GGBS (%) | Silica Fume (%) |
|---|---|---|---|
| $SiO_2$ | 20.5 | 33.8 | 91 |
| IR (insoluble residue) | 0.34 | 0.37 | - |

**Table 2.** *Cont.*

| Chemical Composition | Cement (%) | GGBS (%) | Silica Fume (%) |
|---|---|---|---|
| $Al_2O_3$ | 4.7 | 13.6 | 1.2 |
| $Fe_2O_3$ | 4.0 | 0.7 | 2 |
| CaO | 64.1 | 39.4 | 0.4 |
| MgO | 1.8 | 6.2 | 0.6 |
| $SO_3$ | 2.4 | 0.09 | - |
| S | - | 0.92 | - |
| $Na_2O$ | 0.58 | 0.46 | 1.1 |
| $Mn_2O_3$ | - | 0.27 | - |
| LOI (Loss on Ignition) | 1.5 | 2.0 | 2.9 |
| $Cl^-$ | 0.02 | 0.01 | - |
| $C_3A$ | 5.7 | - | - |

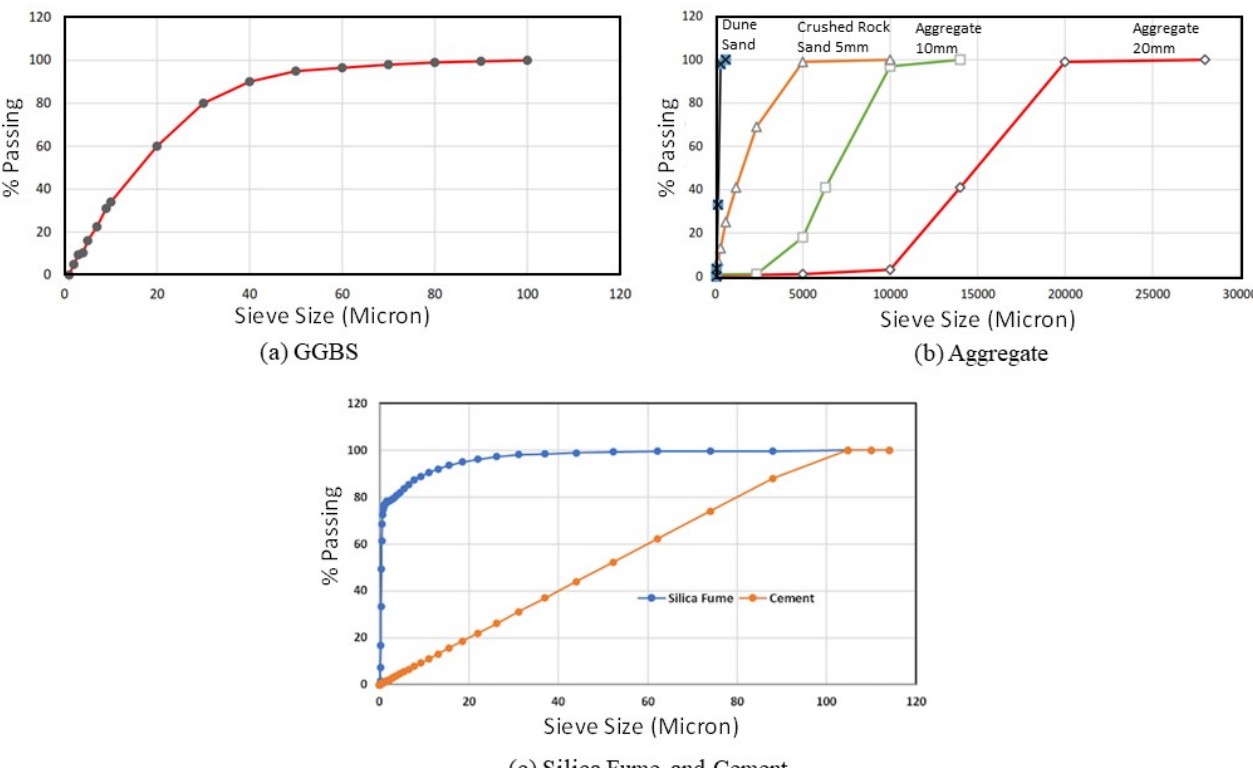

**Figure 1.** PSD of the GGBS, aggregate, silica fume and cement.

### 4.2. Particle Packing by EMMA

The blend design in this study was accomplished using the EMMA program, which is based on the work of Andreassen and Andersen [35], Myhre [36], and German [37]. In the Andreassen and Andersen theoretical model, which is commonly referred to as the Andreassen model, the authors noted that for performing beat compaction and air-free blends, optimum packing happens when the particle size distribution can be described by the following equation:

$$CPFT(\%) = \left(\frac{d}{D}\right)^q \cdot 100 \qquad (1)$$

where CPFT is the cumulative (volume) percentage finer than particles with diameter d, d is the particle diameter, D is the larger particle diameter, and q is the distribution coefficient. EMMA also has an option for determining the proper PSD for blends with infinitely small particles using the method developed by Dinger and Funk [38], which is referred to as the modified Andreassen model.

An advantage of the Andreassen model is that it does not consider the minimum particle diameter and assumes a sequence of decreasing particles, enabling the CPFT to equal zero without significant intervention on the material density. To use EMMA, one needs to provide the distribution of particle size (DPS) of every component of the blend, density of the constituent materials, distribution modulus, maximum particle size, quantity of materials, and water content. EMMA will then compute the optimum DPS of the entire blend and compare it to the Andreassen model, which depicts a linear packing relationship between particles passing through specific sieves and the quantity of those particles. For very fine particles, EMMA has an enhanced feature within the model that can be utilized. Note that optimization of mix designs with the help of the EMMA software has successfully been carried out in the past by many researchers [39–45].

In this study, the methodology consists of altering the quantities of materials in cycles until a match between the recipe model and the EMMA software that is based on Andreassen model was found. The DPS of the blend without alternative cementitious materials C500-S0-G0 is presented in Figure 2a, in which the drop in very fine particle sizes is clearly shown. The PSD for blend C320-S50-G180 is seen in Figure 2b, in which GGBS and SF cause significant improvement in packing within the fine and very fine particle region. In the concrete mix designation of Figure 2, the letters C, S, and G refer to cement, silica fume, and GGBS, respectively.

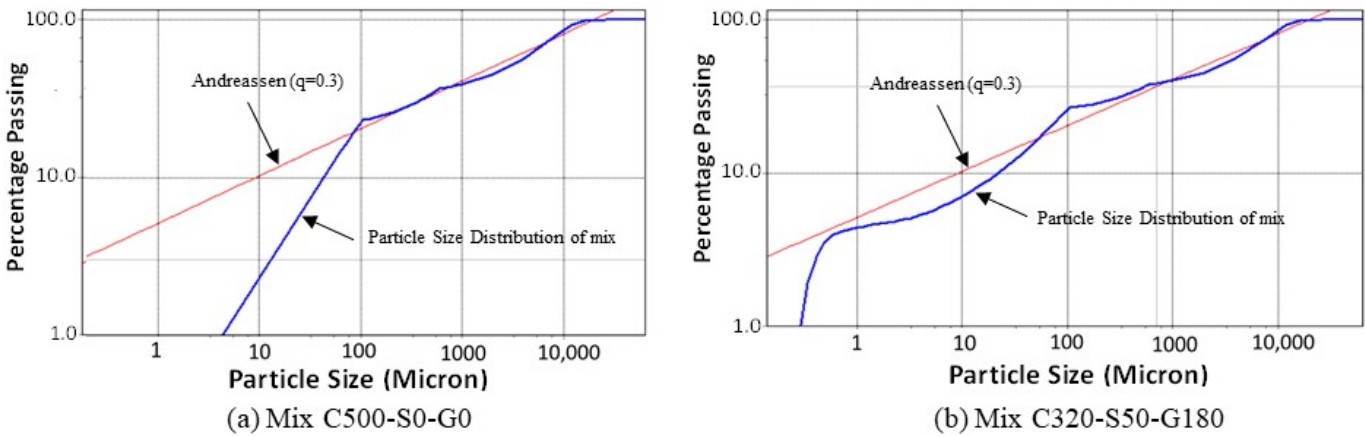

**Figure 2.** PSD modeling of mixes C500-S0-G0 and C320-S50-G180.

### 4.3. Tests Considered

One hundred cubes of 150 × 150 × 150 mm, 9 cylinders of diameter of 150 mm and height of 300 mm, and twenty prisms of dimensions 150 × 150 × 500 mm were cast of different blends. The concrete materials were prepared and then loaded into a ready-mix truck. The mixer rotates the mix at a speed of 12 to 15 rpm.

With regard to the fresh concrete properties, the slump was conducted according to ASTM C143 [46], while the air content test was performed according to ASTM C231 [47]. One day after the casting, the samples were demolded and cured for two periods, 7 and 28 days, at a temperature of about 25 °C. Compressive strength tests of the cubes were also performed at two stages, 7 and 28 days, according to the BS EN12390-2:2009 standard [48], in which the obtained results represent the average strength of three cubes for every blend. The tensile strength of concrete was found by conducting flexural strength tests following the ASTM C293 standard [49]. In such a test, the load was applied at the mid span of a plain concrete prism of dimensions 150 × 150 × 500 mm, where the supports were located 25 mm

from the edge of the prism and the average of two prisms was recorded for every blend. The modulus of elasticity (MOE) of the concrete was assessed by loading a $150 \times 300$ mm cylinder in compression at 28 days with a stress equal to 20% of the cube's compressive strength, noting the deflection at that level of load, and computing the corresponding elastic stiffness.

Testing of resistance to chloride penetration, referred to as the rapid chloride penetration test (RCPT), was performed according to the ASTM C1202 standard [50] by coring $90 \times 150$ mm concrete cylinders from concrete cubes and measuring the electrical connectivity by placing the concrete specimen between two electrical poles. In the test, every electrical pole was represented by a solution reservoir of 0.25 L that had a stainless steel mesh on the inner side adjacent to the specimen, in which one side was filled with a 3.0% NaCl solution by weight, while the other side was filled with a 0.3 molar concentration of NaOH. The system was subjected to 60 volts of electrical direct current potential for 6 h, and the passing electrical current within the samples was measured. The coulomb values can be found by the area under the curve of current versus time and were compared to the ASTM C1202 limits presented in Table 3 to identify the level of permeability.

**Table 3.** Chloride permeability limits [51].

| Chloride Permeability | Charge (Coulombs) | Type of Concrete | Total Integral Chloride to 41 mm Depth after 90-Day Ponding Test |
|---|---|---|---|
| High | >4000 | High water-to-cement ratio (>0.6) conventional Portland cement concrete | >1.3 |
| Moderate | 2000–4000 | Moderate water-to-cement ratio (0.4–0.5) conventional Portland cement concrete | 0.8–1.3 |
| Low | 1000–2000 | Low water-to-cement ratio (<0.4) conventional Portland cement concrete | 0.55–0.8 |
| Very low | 100–1000 | Latex-modified concrete, internally sealed concrete | 0.35–0.55 |
| Negligible | <100 | Polymer-impregnated concrete, polymer concrete | <0.35 |

After casting and curing the cubes for 7 and 28 days, selected samples were taken for the SEM analysis to study the microstructure and morphology of the concrete. Specimens containing different components of concrete were cut from the center of the cubes to be representative. A piece of concrete was cut in a regular shape with a diameter of around 10 mm and 5 mm thickness with a flat surface obtained by using a regular grinder with a concrete-cutting desk. To obtain clear images, the specimens were polished to obtain a smooth surface and vacuumed in a chamber. The specimens were then placed in a special salver and fixed by screws to prevent any movement while being tested inside the SEM. After trials, the backscattered mode was selected to obtain the micrograph images and to analyze the microstructure of the specimens. Chemical composition analysis of areas in the concrete specimens was undertaken using the X-ray analysis feature in the SEM. A summary of the tests on the fresh and hardened concrete considered in this study is presented in Figure 3.

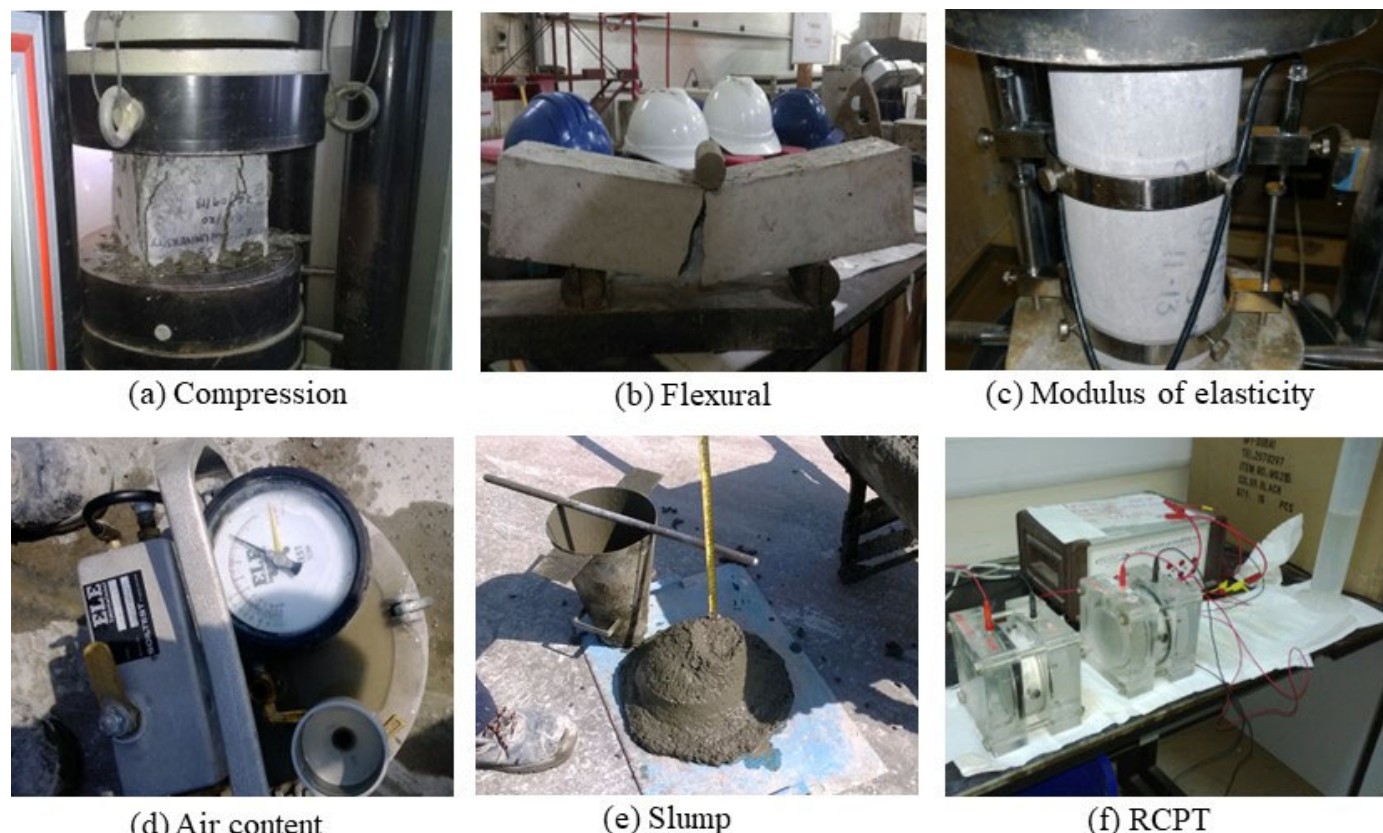

**Figure 3.** Summary of tests conducted on fresh and hardened concrete.

## 5. Results and Discussion

### 5.1. Modeling Analysis by EMMA Program

Blends from previously published research [52] were modeled in this study, and the proportions were adjusted using the EMMA program. Table 4 presents the proportions for two control conventional concrete mixes with target cubic strengths of 40 and 60 MPa, denoted by C40 and C60, respectively. The findings of numerous trials by the EMMA program led to the blends presented in Table 5, which included one mix without and seven mixes with supplementary cementitious materials. The eight mix designs of the green concrete depict the optimum match between the model and the real particle size distribution of the materials. Superplasticizer was used in similar amounts for all the mixes.

**Table 4.** Mix proportions of 40 MPa and 60 MPa without supplementary materials.

| Material | 20 mm Aggregate (kg) | 10 mm Aggregate (kg) | 5 mm Crushed Rock (kg) | Dune Sand (kg) | Cement (kg) | w/c Ratio |
|---|---|---|---|---|---|---|
| C40 | 720 | 360 | 520 | 220 | 410 | 0.3 |
| C60 | 720 | 390 | 540 | 220 | 500 | 0.3 |

Although the PSD of the lower-strength control mix C40 blend in Figure 4a shows a well-graded trend close to the Andreassen model, an area of enhancement in particles packing can be clearly recognized in the graph. Cement and GGBS area can be enhanced greatly by adding smaller particles to fill the gap in the area of 1 to 10 microns. Additionally, it can be recognized from the graph that a small extra amount of 10 mm aggregate can be added to minimize the drop and match the model within that region of the graph to have better grading. Figure 4b shows the C60 mix that lacks very fine particles as shown in the highlighted circular part of the graph. Improvement in the particle distribution for the higher-strength control C60 mix, denoted C500-S0-G0 and presented earlier in Figure 2a, is

accomplished by addressing the fine and very fine particles within the region between 100 and 10,000 microns to match the Andreassen model, while keeping material proportions elsewhere the same.

**Table 5.** Mix proportions adjusted by EMMA.

| Material (kg) | C500-S0-G0 | C320-S0-G180 | C500-S100-G0 | C320-S50-G180 | C250-S0-G250 | C275-S0-G275 | C250-S50-G250 | C500-S50-G0 |
|---|---|---|---|---|---|---|---|---|
| 20 mm Aggregate | 100 | 100 | 100 | 100 | 100 | 100 | 100 | 100 |
| 10 mm Aggregate | 810 | 810 | 810 | 950 | 810 | 810 | 950 | 810 |
| 5 mm crushed rock | 580 | 580 | 580 | 580 | 580 | 580 | 580 | 580 |
| Dune sand | 280 | 280 | 250 | 250 | 280 | 250 | 250 | 250 |
| Cement | 500 | 320 | 500 | 320 | 250 | 275 | 250 | 500 |
| GGBS | 0 | 180 | 0 | 180 | 250 | 275 | 250 | 0 |
| Silica fume | 0 | 0 | 100 | 50 | 0 | 0 | 50 | 50 |
| W/C ratio | 0.3 | 0.3 | 0.3 | 0.3 | 0.3 | 0.3 | 0.3 | 0.3 |

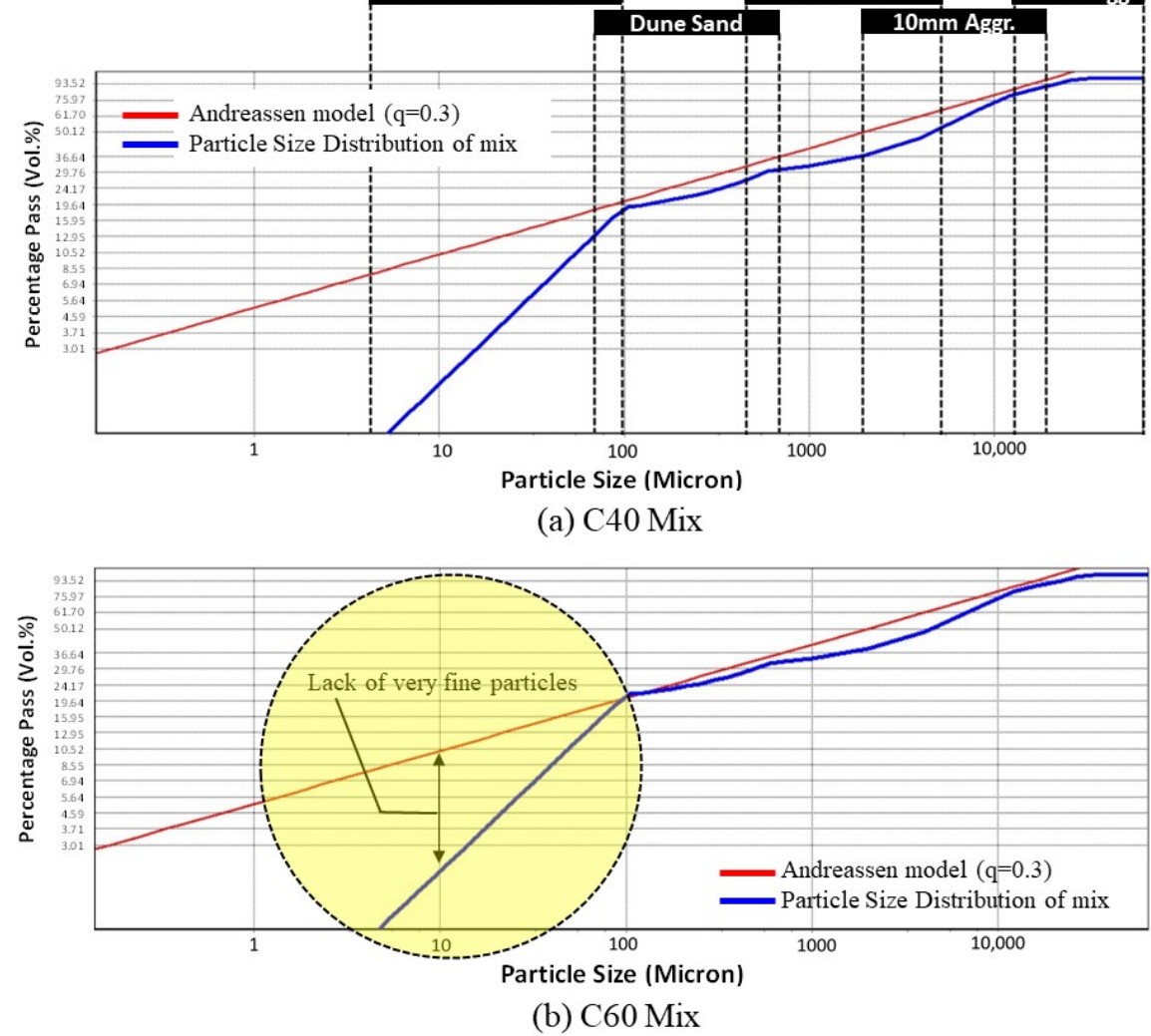

**Figure 4.** Features of PSD of the control C40 and C60 concrete mixes.

For mix C320-S0-G180, shown in Figure 5a, there is no change in the obtained graph, as the PSD of cement and GGBS is the same. Note that adding SF into the mix results in C500-S100-G0, for which the PSD is presented in Figure 5b, leads to enhancement in the

particle grading, since SF can fill the area from 0.5 to 65 microns. The lesser SF in mix C320-S50-G180 leads to relatively inferior grading of the mix, as shown in Figure 5c. Overall, it can be seen from the figures that adding SF can greatly improve the particle grading when it is compared to mixes without SF, since the presence of such small particles fills microgaps in the concrete's microstructure. The mixes that include 50% cement replacement with GGBS in the blends are shown in Figure 5d–f. Additionally, SF has been added to the mixes that are denoted C250-S50-G250 and C500-S50-G0, and their PSD graphs are presented in Figure 5f,g. Note that the Andreassen model is represented by the straight red line in all graphs of Figure 5 and is based on distribution coefficient q = 0.30. The value of q has been chosen based on the type of concrete used in the study, which was either self-flowing or conventional. According to EMMA's software manual, self-flowing concrete q value is 0.25 or below, while higher values can be considered to achieve a conventional concrete. A value of q = 0.3 was chosen as a constant value for the packing of all the mixes for self-flowing concrete.

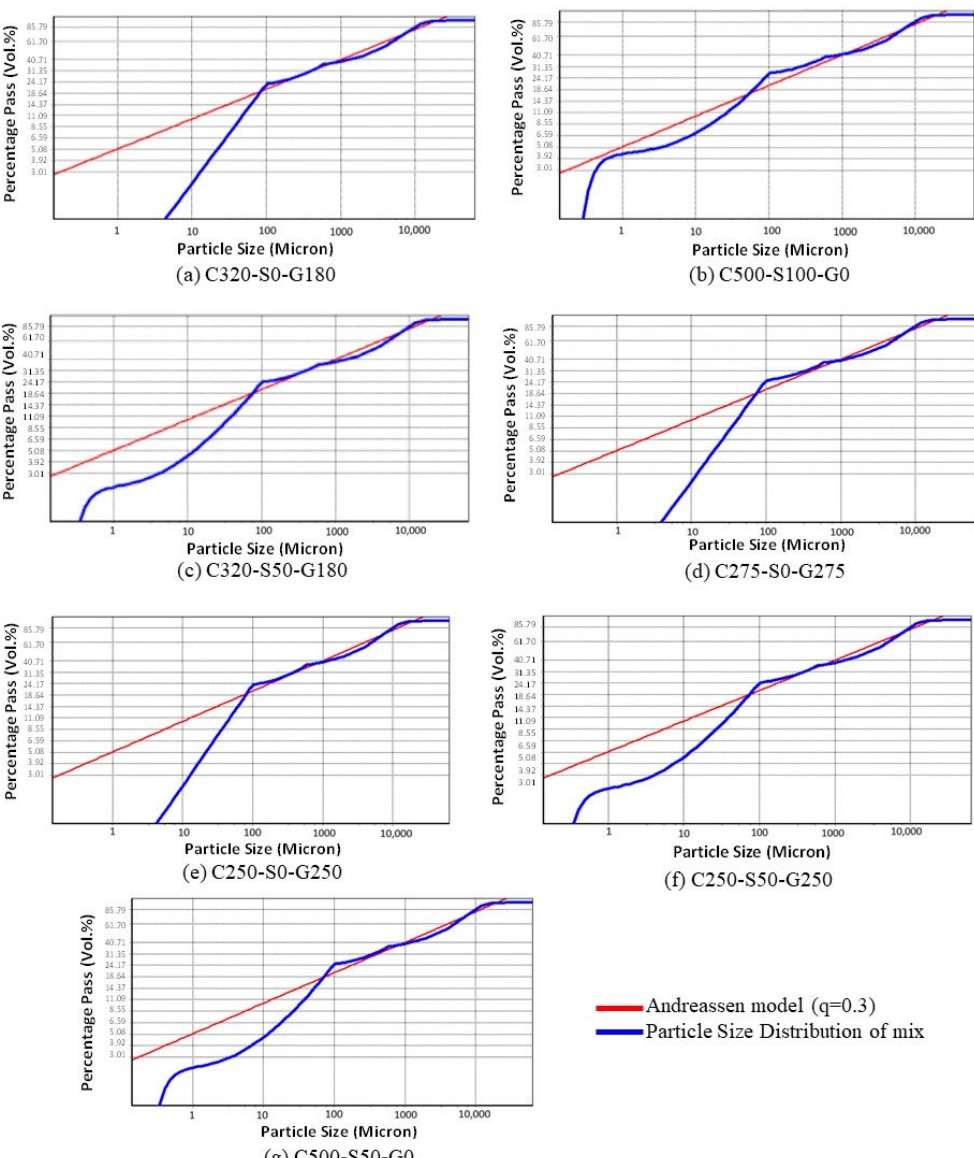

**Figure 5.** PSD of the considered mixes and comparison with theoretical model.

### 5.2. Fresh Concrete Properties

Slump, temperature, air content, and dry density tests were performed on the fresh concrete of the mixes, as shown in Table 6. In general, the slump of the green concrete ranged between 122 and 168 mm, due mainly to the effect of the supplementary cementitious materials, such as GGBS, which produced higher rheological effects due to their lubrication effects and lower hydration rate than Portland cement. However, mixes with higher SF exhibited small loss of workability as a result of the higher hydration due to the presence of SF. The temperature of the green concrete was very close to that of traditional concrete, varying within a narrow range (28.9–32.0 °C). Air content test findings showed very low air values, within the range of 1.0–1.4%, in all the new particle-packed mixes compared to conventional C40 and C60 mixes, and the published literature on the subject, wherein normal air content varies from 1.4% to 2% [53]. In general, the density of the GGBS and/or SF concrete cubes was between 2477 and 2551 kg/m$^3$, well below the corresponding density of the C60 traditional mix, which contained a larger quantity of 20 mm aggregate (2604 kg/m$^3$).

**Table 6.** Slump, temperature, air content and dry density test results.

| Ref. Mix | Slump (cm) | Temperature (°C) | Air Content (%) | Average Density of Cubes (kg/m$^3$) | Average Density of Prisms (kg/m$^3$) |
|---|---|---|---|---|---|
| C40 | 12.8 | 31.2 | 1.9 | 2533 | 2557 |
| C60 | 15.1 | 30.5 | 1.6 | 2604 | 2608 |
| C500-S0-G0 | 12.7 | 32.0 | 1.3 | 2510 | 2655 |
| C320-S0-G180 | 12.2 | 31.0 | 1.0 | 2551 | 2565 |
| C500-S100-G0 | 16.8 | 29.0 | 1.1 | 2495 | 2495 |
| C320-S50-G180 | 14.2 | 30.2 | 1.1 | 2477 | 2487 |
| C250-S0-G250 | 13.5 | 31.6 | 1.4 | 2542 | 2735 |
| C275-S0-G275 | 15.3 | 28.9 | 1.2 | 2536 | 2545 |
| C250-S50-G250 | 14.2 | 30.1 | 1.3 | 2507 | 2513 |
| C500-S50-G0 | 15.4 | 29.8 | 1.3 | 2524 | 2597 |

### 5.3. Compressive Strength

The cube test was conducted according to the relevant British standard, and the values of compressive strength at 7 and 28 days are reported in Table 7 and Figure 6. Figure 7a shows that the failure mode of a cube conformed to the relevant BS standard, which indicates proper test conditions. Also provided in the table is information about the modulus of rupture and modulus of elasticity of the concrete, which will be discussed in Sections 5.5 and 5.6. The results indicate that even though the eight high-performance concrete mixes developed possessed superior particle packing, their compressive strength did not reach the corresponding strength of the conventional C60 mix (67.0 MPa). Cube test results of the green concrete indicated that the compressive strength was at least 50.13 MPa at 7 days and 56.47 MPa at 28 days, which was observed for mix C275-S0-G275. On the other hand, the maximum 7- and 28-day compressive strengths for the concrete that employed alternative cementitious material were 59.47 MPa and 68.09 MPa, respectively, for mix C500-S100-G0. The ratio of the 7- to 28-day compressive strength for the green concrete varied between 0.79 and 0.93. Only one of the seven concrete mixes that utilized alternative cementitious materials (mix C275-S0-G275) did not reach the strength of C500-S0-G0 at the age of 28 days, missing the mark by less than 0.8%, which is considered reasonable. Overall, the seven green concrete mixes had an average 28-day compressive strength of 63.19 MPa, 11% higher than the corresponding strength of C500-S0-G0. Figure 7b shows the shear failure through the aggregate due to the compressive load on the cube. From a

practical sustainability point of view, it can be concluded that replacing 50% of the cement with GGBS and 10% of the cement with SF gives reasonable compressive strength.

**Table 7.** Test results of compressive strength, modulus of rupture and modulus of elasticity.

| Ref. Mix | Compressive Strength (MPa) | | Modulus of Rupture (MPa) | | Modulus of Elasticity (GPa) | |
|---|---|---|---|---|---|---|
| | 7 days | 28 days | Experimental | ACI 318 | Experimental | ACI 318 |
| C40 | 29.01 | 42.00 | 5.5 | 3.21 | 30.46 | 28.62 |
| C60 | 59.61 | 67.00 | 8.5 | 4.06 | 38.47 | 37.46 |
| C500-S0-G0 | 51.30 | 56.92 | 10.5 | 3.74 | 35.46 | 34.06 |
| C320-S0-G180 | 54.37 | 65.85 | 11.0 | 4.02 | 38.14 | 36.11 |
| C500-S100-G0 | 59.47 | 68.09 | 12.0 | 4.09 | 38.78 | 35.38 |
| C320-S50-G180 | 53.26 | 57.31 | 10.5 | 3.75 | 35.58 | 32.2 |
| C250-S0-G250 | 50.62 | 62.30 | 12.5 | 3.91 | 37.10 | 36.8 |
| C275-S0-G275 | 50.13 | 56.47 | 11.5 | 3.73 | 35.32 | 33.1 |
| C250-S50-G250 | 53.91 | 66.63 | 9.0 | 4.05 | 38.36 | 35.31 |
| C500-S50-G0 | 51.93 | 65.66 | 9.0 | 4.02 | 38.08 | 36.12 |

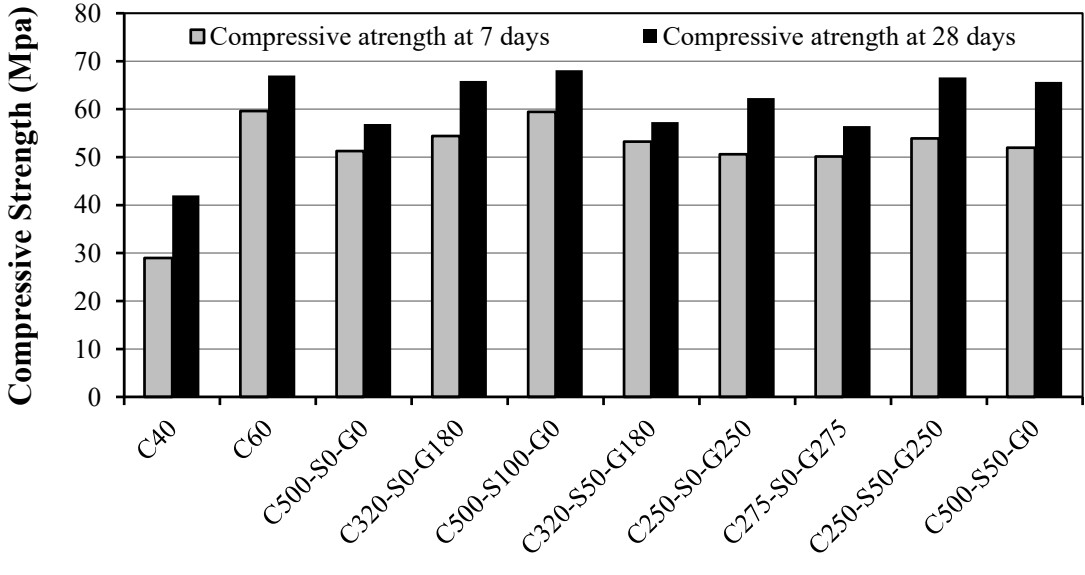

**Figure 6.** Compressive strength of concrete at 7 and 28 days for all mixes.

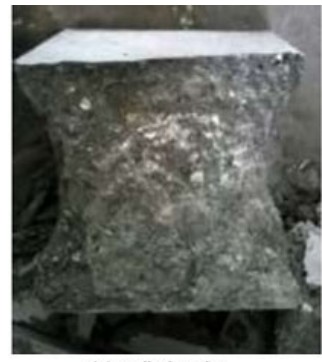
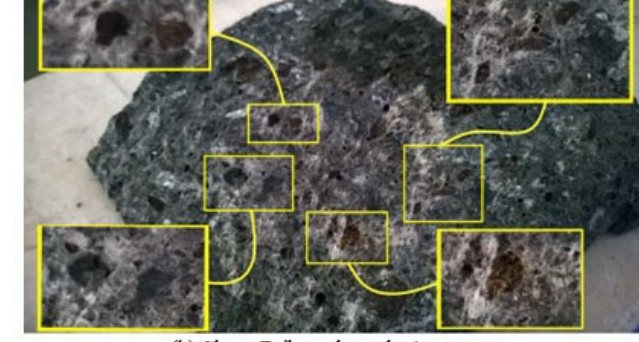

(a) Failed cube

(b) Shear Failure through Aggregate

**Figure 7.** Failure mode of green concrete under compressive stress.

### 5.4. Permeability

The findings of tests for resistance to chloride penetration at 28 days are shown in Table 8. The results indicate that all eco-friendly concrete blends have less permeability than C40 and C60. In reference to Table 3 [51], the readings of RCPT are between low and very low for almost all the considered environmentally friendly mixes. Figure 8 shows the findings of RCPT and air content on the same graph. Compared to conventional concrete, the mixes with supplementary cementitious materials exhibit typically lower values of RCPT and air content. In general, the RCPT results are higher when the mixes have higher air content. Note that concrete mix C320-S0-G180 showed the lowest air content and porosity among all the considered mixes.

**Table 8.** RCPT results for the considered concrete at 28 days.

| Ref. Mix | RCPT (Coulombs) | RCPT Reading |
|---|---|---|
| C40 | 6451 | High |
| C60 | 3189 | Moderate |
| C500-S0-G0 | 2744 | Moderate |
| C320-S0-G180 | 392 | Very low |
| C500-S100-G0 | 1856 | Low |
| C320-S50-G180 | 684 | Very low |
| C250-S0-G250 | 1553 | Low |
| C275-S0-G275 | 2517 | Moderate |
| C250-S50-G250 | 904 | Very low |
| C500-S50-G0 | 615 | Very low |

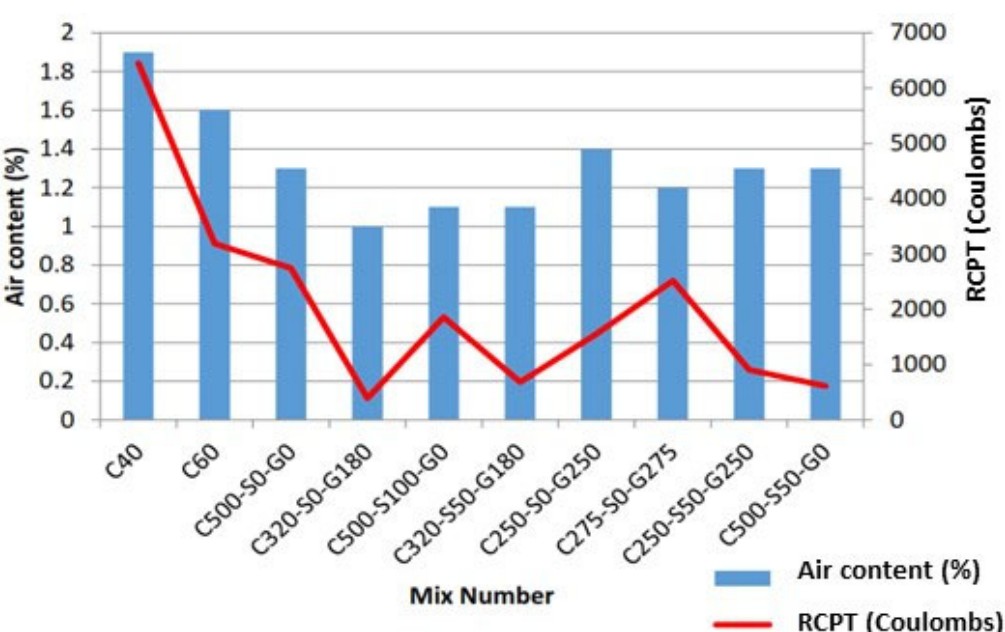

**Figure 8.** RCPT and air content results for the concrete mixes.

### 5.5. Flexural Strength

Table 7 shows the results of flexural strength as determined by the modulus of rupture (MOR) tests, also referred to as $f_r$, which is considered an indicator of the required tensile stress to initiate the first crack in concrete. All eco-friendly concrete mixes showed a higher modulus of rupture than normal concrete by 5–30%, as seen in Figure 9. Table 6 presents

the density of the concrete prisms, which ranged between 2487 and 2735 kg/m$^3$. This difference in density represents the randomization of particle distribution in every mix. In general, concrete specimens with higher weight density possess more fine particles (as represented by the binder material) and less aggregate than lighter ones, which contributed towards their higher flexural strength. Comparison of the MOR experimental results with the theoretical values from the ACI318-19 code [54], given by $f_r = 0.62\sqrt{f_c'}$ where f'$_c$ (MPa) is the concrete cylinder strength at 28 days taken as 80% of the cube strength [55], indicate that the code is very conservative in its prediction of the MOR. For the eight optimally packed concrete mixes, the code underestimates the flexural strength by 55–69%. This finding is in line with published research on the subject [56].

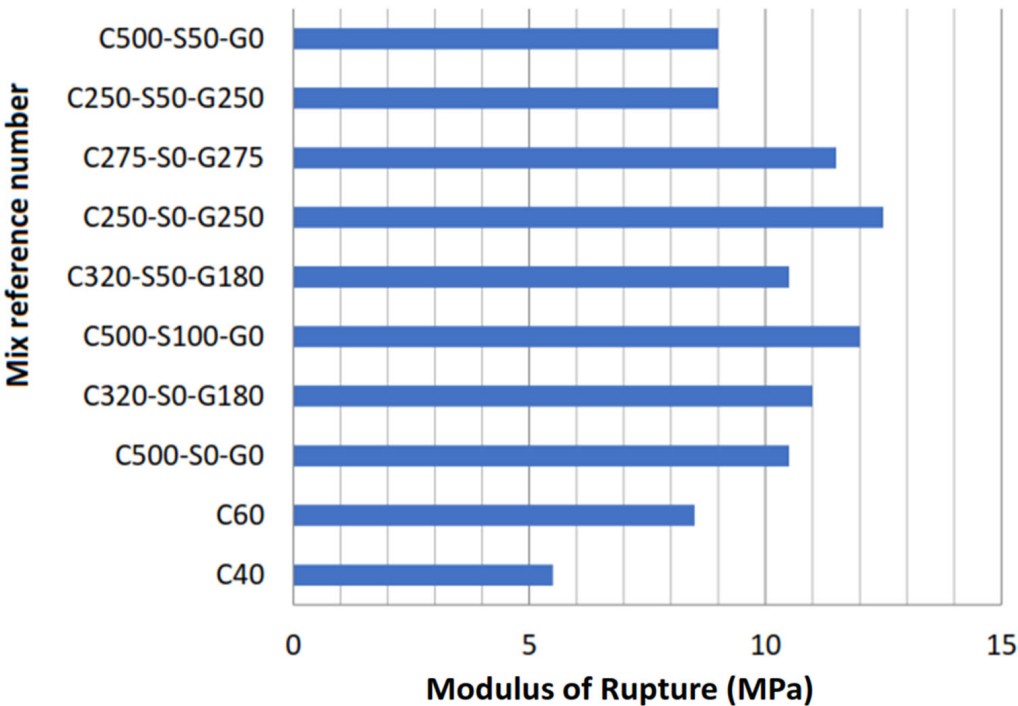

**Figure 9.** Modulus of rupture results for the concrete mixes.

*5.6. Modulus of Elasticity*

In this study, the initial modulus of elasticity (MOE) of the concrete from the experiments is determined by averaging the slopes of the stress–strain relationship of a cylindrical concrete specimen under compression after loading and reloading it up to 40% of the compressive strength. As expected, Figure 10 of the experimental stress–strain relationship for C250-S50-G250 confirms that it is linear during the two loading and reloading stages, leading to MOE equal to 38.36 GPa. Table 7 shows the results of the MOE of all the considered concrete, which indicate that the MOE is not greatly influenced by the addition of GGBS or SF. Overall, the experimentally obtained MOE values are very close to the theoretically predicted ones by the ACI 318-19 code [54], given by $E_c = 0.043(w_c)^{1.5}\sqrt{f_c'}$ where w$_c$ = mass density of the concrete (kg/m$^3$), provided in Table 6. The difference between the experimental and predicted values for the eight optimally packed concrete mixes ranges between 0.8% and 8.8%.

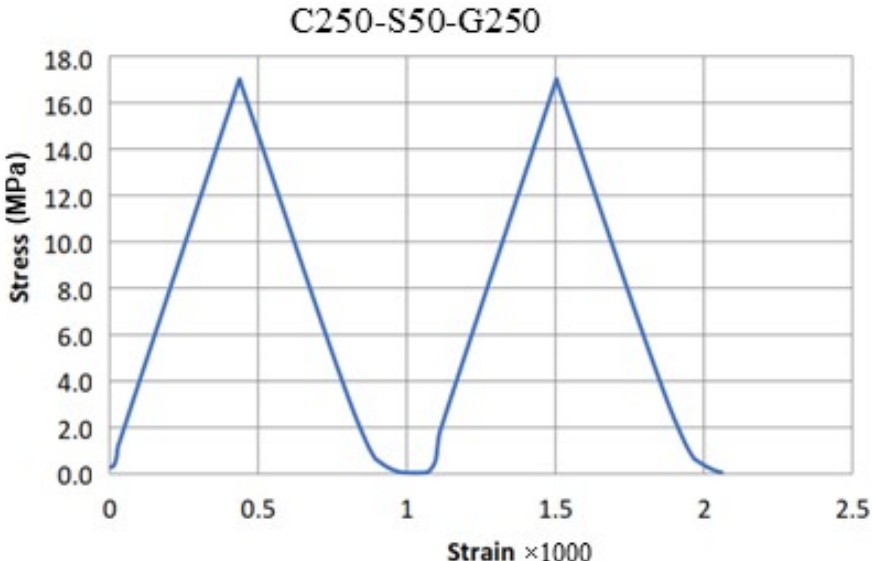

**Figure 10.** Modulus of elasticity by loading and unloading of C250-S50-G250.

*5.7. Microstructural Analysis*

Three distinct samples were tested using the SEM in order to examine the morphology of the concrete: one without supplementary materials (C500-S0-G0), one with GGBS (C320-S0-G180) and one with SF (C500-S100-G0), as presented in Figure 11. Figure 11a shows the microstructure of concrete using an SEM image of mix C500-S0-G0 where ettringite was detected. Ettringite is formed due to the reaction between calcium aluminate and calcium sulfate, which results in voids that are relatively big and deep, causing an increase in the porosity of concrete and producing open microstructure. Open microstructure also shows cave-shaped voids of size 2.5 to 5 microns in diameter at locations close to the ettringite formations. Calcium hydroxide (CH), which appears as crystal shells covered with C-S-H gel, is one of the main products of the hydration process, although its contribution to concrete strength is minimal. Loose particles, voids and microcracks of different widths and lengths were formed due to lack of C-S-H gel formation. The existence of calcium hydroxide (CH), ettringites, and voids is an indication of concrete weakness. The sample that contains GGBS from the mix C320-S0-G180 that was observed under the SEM is shown in Figure 11b. Compared to Figure 11a, the C320-S0-G180 mix shows a much better microstructure due to the presence of fewer voids, smaller cracks and a more homogeneous structure. These findings explain the better performance of concrete containing GGBS in RCPT, where the microstructure of the concrete containing GGBS demonstrates fewer and smaller voids and cracks than the regular concrete mix. The SEM examination for a sample of mix C500-S100-G0, which contains SF, is shown in Figure 11c. The results indicate very small and discontinued voided areas due to a very dense microstructure, and no ettringite was observed. Although some images exhibited cracks of 2 microns long or less, they generally had negligible depth due to the high density of the mix. The observed voids were also very small in size and frequency. Therefore, RCPT results of mix C500-S100-G0, which indicated very low permeability, can be justified. The smaller voids and denser microstructure make it hard for any fluid to penetrate through concrete. Note that earlier studies have shown that SF addition decreases the ratio of Ca/Si of C-S-H in concrete, where in some cases it reaches 1.0 [57].

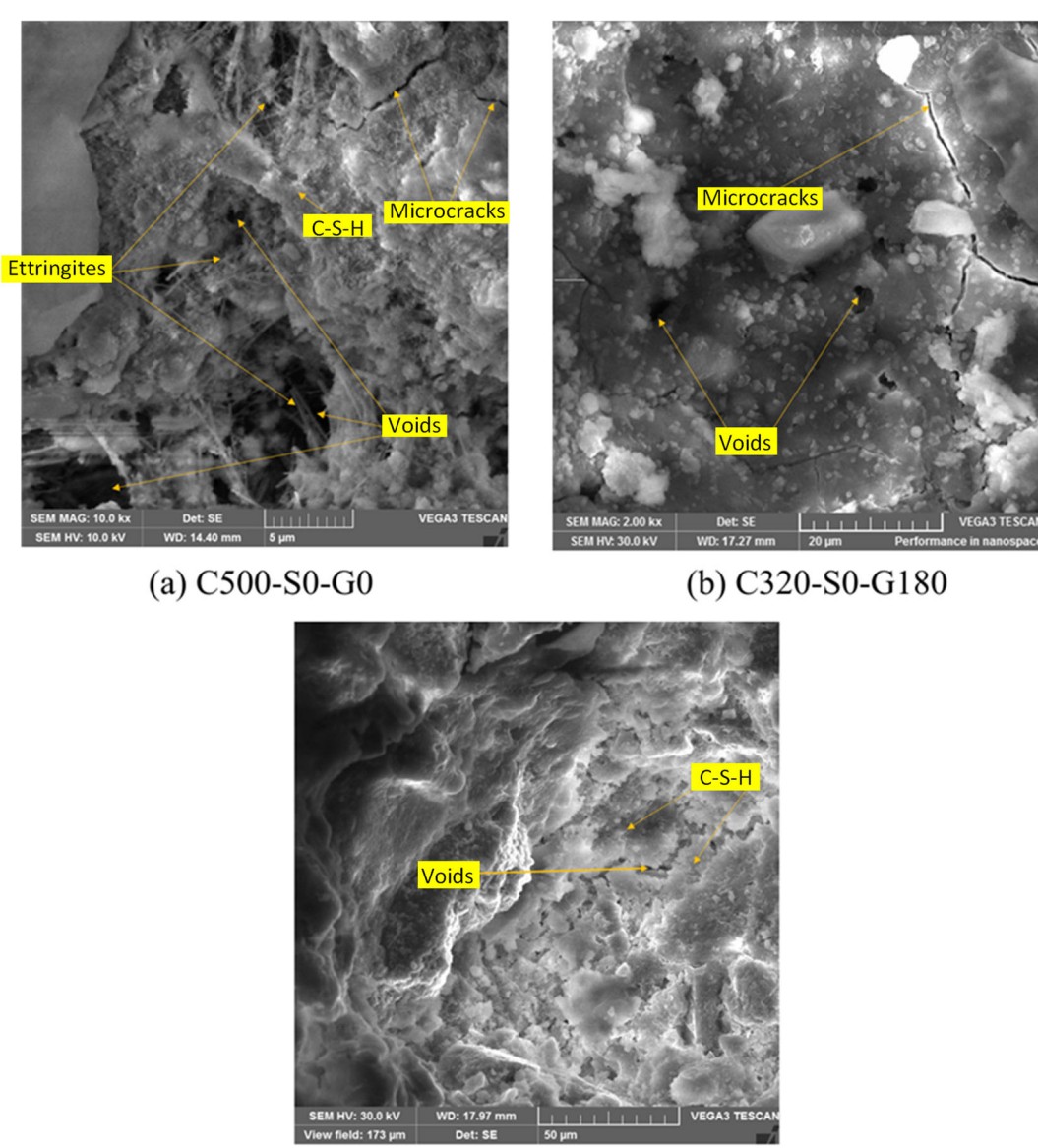

**Figure 11.** Image of concrete microstructure using secondary electron mode.

*5.8. Chemical Composition*

With the help of a scanning electron microscope, a chemical composition analysis that was accomplished by energy-dispersive spectroscopy (EDS) was carried out on the concrete that employed GGBS and SF.

Figure 12 shows the analysis of chemical components for mix C320-S0-G180 in four spectra. While the overall results detect many chemical components within the considered spectra, C-S-H appears to be the main component in the presence of calcium. The ratios between the quantity of calcium (Ca) and silicon (Si) confirm good hydration of the cementitious material. Analysis of Spectrum 11 in Figure 12b shows the presence of calcium hydroxide ($Ca(OH)_2$), which is the main component of the cement hydration process. The importance of the presence of $Ca(OH)_2$ in the mix is that GGBS particles will react with it to create C-S-H gel. With time, concrete keeps gaining strength and enhancement in durability throughout the formation of C-S-H. Chemical composition of Spectrum 12 in Figure 12c, Spectrum 13 in Figure 12d and Spectrum 15 in Figure 12e confirms the presence of C-S-H gel, since the Ca/Si ratios in these three spectra are equal to 1.5, 1.8 and 1.9, respectively.

Although the dense microstructure for the mix containing GGBS shows visible cracks and voids, they are scattered, small and shallow.

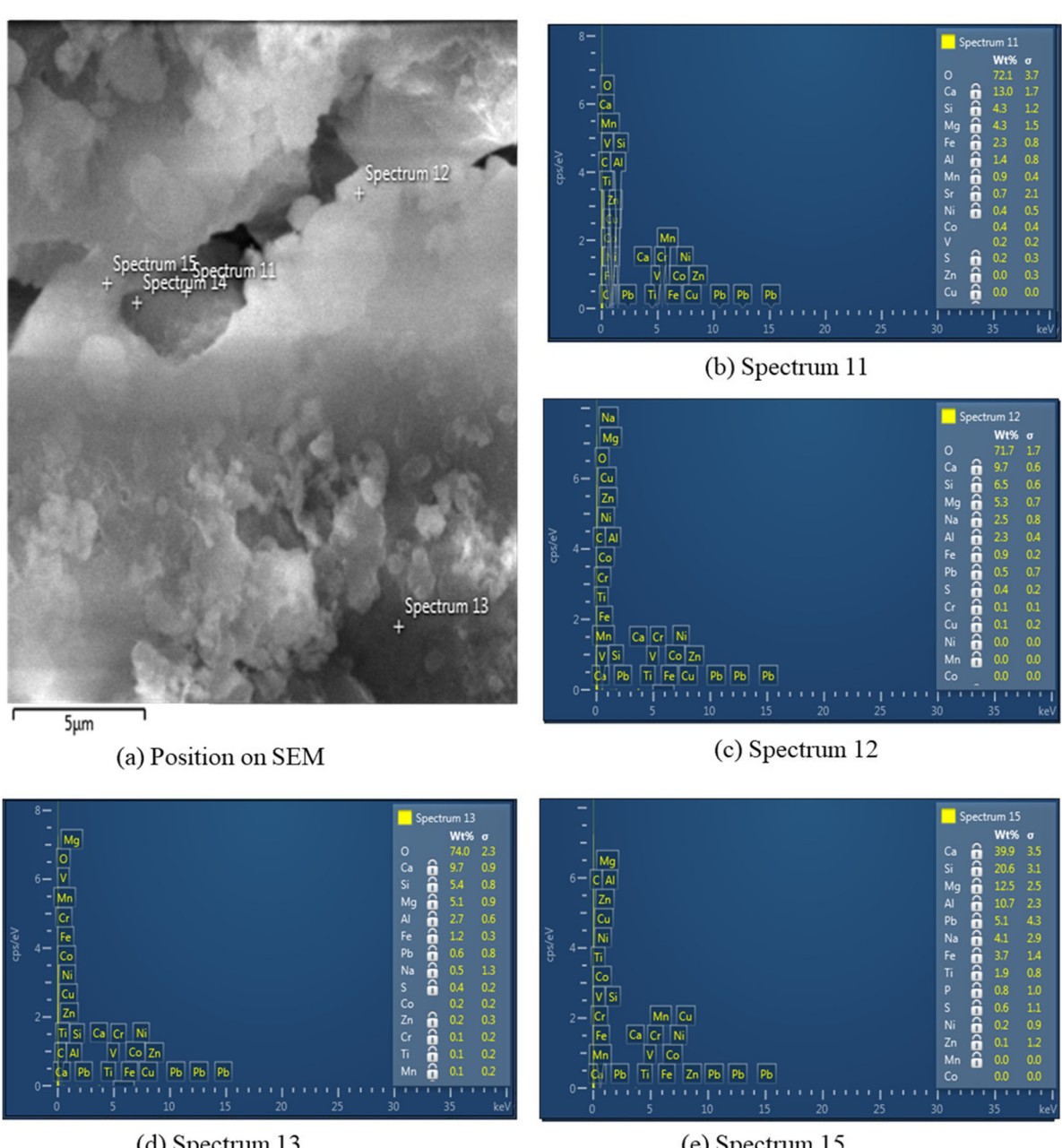

**Figure 12.** Analysis of chemical components for mix C320-S0-G180.

The small voids, dense microstructure and discontinuity in the cracks in the C500-S100-G0 mix make it hard for any fluid to penetrate through concrete. The addition of SF decreases the ratio of Ca/Si, resulting in C-S-H within the concrete, where in some cases it reaches less than 1.0, as shown in Figure 13. Analysis of the chemical components of the concrete shows that magnesium also reacts with SF in some cases in a similar manner to CaO, as shown in Figure 13b for Spectrum 68. The high percentage of Si combined with a low percentage of calcium and magnesium in Spectrum 69, presented in Figure 13c, represents sand particles. Magnesium silicate hydrate (M-S-H) is represented by Spectrum 70, shown in Figure 13d. M-S-H gel is identified when the Mg/Si ratio falls in the range of 0.7 to 1.5 [53]. M-S-H is the main hydration product and was only found with specimens that had SF. In Spectrum 71 (Figure 13e), the Ca/Si ratio is 0.9, which also means this

spectrum represents C-S-H gel. Enhanced layers of concrete microstructure can be seen in SEM images that contain SF, which contributes to better packing and fewer voids.

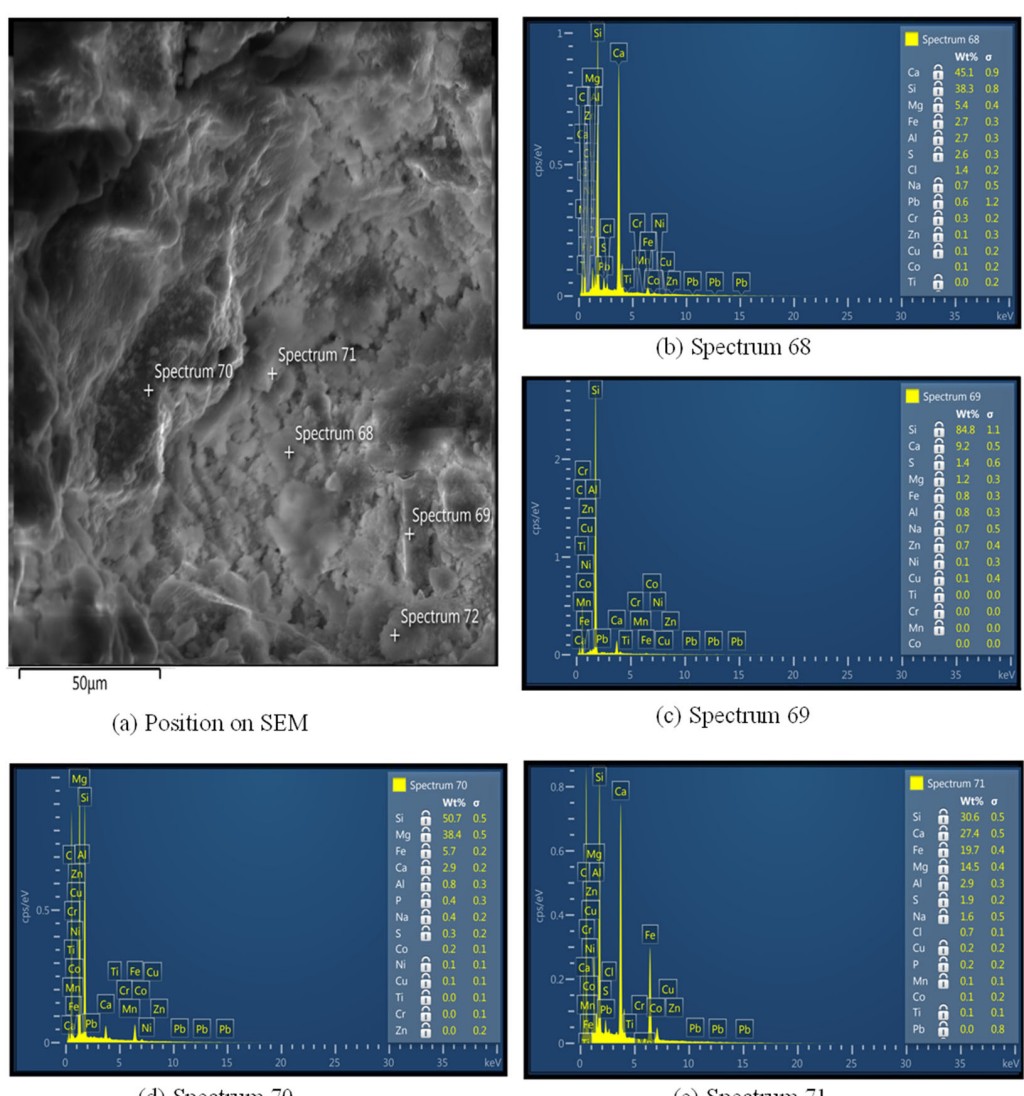

**Figure 13.** Analysis of chemical components for mix C500-S100-G0.

## 5.9. Cost Analysis and Environmental Impact

The prices of various components of concrete in the UAE markets are reported in Table 9, with the transportation of material included within these prices. Depending on the components of each mix, the price per cubic meter for each blend was computed as seen in Figure 14. Additionally, the cost of all materials utilized in each mix was calculated and is summarized in Table 10. It can be seen that replacing cement with GGBS did not add to cost of the production of the eco-friendly concrete in UAE. Although replacing cement by 10% of the cement with SF can greatly improve the permeability of the concrete in early age and beyond [52], this is achieved at only a 13% increase in the cost of the product based on the UAE market. Based on the findings of the study, mix C320-S0-G180 gave the best results at lower cost than conventional concrete.

**Table 9.** Cost of individual constituents of concrete in the UAE.

| Material | 20 mm Agg. (m³) | 10 mm Agg. (m³) | 5 mm Crushed Rock (m³) | Dune Sand (m³) | Cement (ton) | GGBS (ton) | Silica Fume (ton) | Water (m³) | Super-Plasticizer (milliliter) |
|---|---|---|---|---|---|---|---|---|---|
| Cost (AED/unit) | 29.5 | 29.5 | 30 | 22 | 220 | 225 | 1,000 | 10.6 | 1.5 |

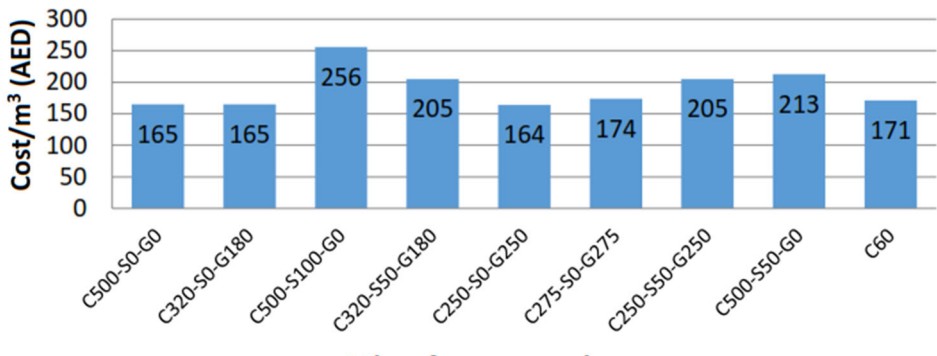

**Figure 14.** Cost per cubic meter for the considered mixes.

**Table 10.** Cost matrix for normal and green concrete in the UAE.

| Material \ Ref. Mix | C60 | C500-S0-G0 | C320-S0-G180 | C500-S100-G0 | C320-S50-G180 | C250-S0-G250 | C275-S0-G275 | C250-S50-G250 | C500-S50-G0 |
|---|---|---|---|---|---|---|---|---|---|
| 20 mm Aggregate | 18.85 | 1.74 | 1.74 | 1.74 | 1.74 | 1.74 | 1.74 | 1.74 | 1.74 |
| 10 mm Aggregate | 12.56 | 14.06 | 14.06 | 14.06 | 16.49 | 14.06 | 14.06 | 16.49 | 14.06 |
| 5 mm crushed rock | 0.00 | 9.94 | 9.94 | 9.94 | 9.94 | 9.94 | 9.94 | 9.94 | 9.94 |
| Dune Sand | 0.00 | 3.42 | 3.42 | 3.06 | 3.06 | 3.42 | 3.06 | 3.06 | 3.06 |
| Cement | 110.0 | 110.0 | 70.4 | 110.0 | 70.4 | 55.0 | 60.5 | 55.0 | 110.0 |
| GGBS | 0.00 | 0.00 | 40.50 | 0.00 | 40.50 | 56.25 | 61.88 | 56.25 | 0.00 |
| Silica fume | 0.00 | 0.00 | 0.00 | 100.00 | 50.00 | 0.00 | 0.00 | 50.00 | 50.00 |
| water | 1.48 | 1.48 | 1.48 | 1.77 | 1.63 | 1.48 | 1.63 | 1.63 | 1.63 |
| Super-plasticizer | 15.0 | 15.0 | 15.0 | 18.0 | 16.5 | 15.0 | 16.5 | 16.5 | 16.5 |
| Material cost/Mix (AED) | 158 | 156 | 157 | 259 | 210 | 157 | 169 | 211 | 207 |
| Cost/m³ (AED) | 171 | 165 | 165 | 256 | 205 | 164 | 174 | 205 | 213 |

Decreasing the quantity of cement by replacing it with GGBS can reduce the carbon dioxide ($CO_2$) emissions in the production of eco-friendly concrete, as shown in Figure 15. For mixes C250-S0-G250 and C250-S50-G250, the estimated $CO_2$ emission was 50% less than that of conventional concrete (C60). Additionally, the heat produced from the chemical reaction between cement and water in the hydration process can be reduced when the cement is replaced with the same ratio of a substitute by the waste of industrial products, such as GGBS and SF. When the heat of the hydration process is reduced, internal stresses in concrete can decrease, which is advantageous because it leads to reduction in the possible formation of early-age shrinkage cracks.

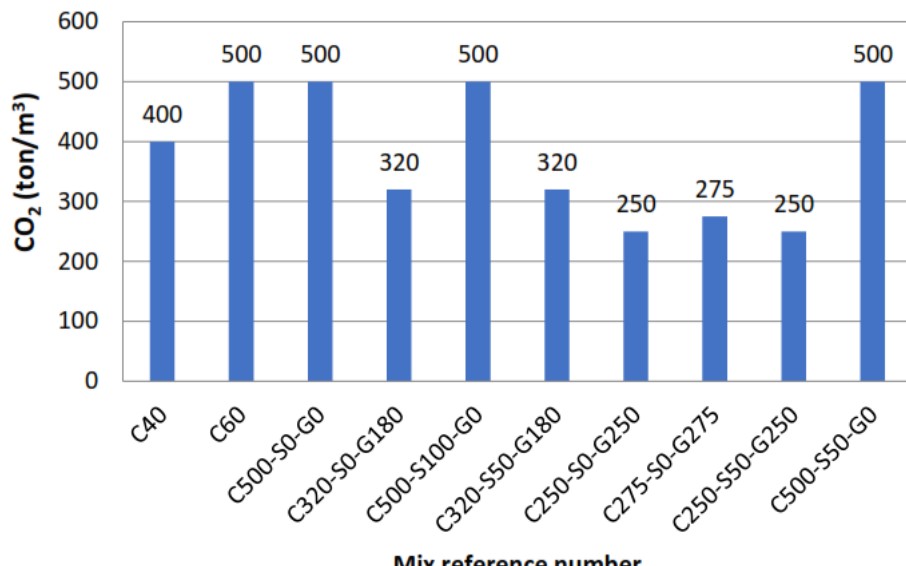

**Figure 15.** Carbon dioxide emissions (ton/m$^3$) for the considered mixes.

## 6. Conclusions

This study demonstrated the feasibility and cost-effectiveness of developing sustainable concrete mixes that have lower environmental impact and higher durability than conventional concrete. The optimal mix of sustainable concrete was obtained with the help of a tool that considered the packing density of concrete materials. The packing density results were verified using scanning electron microscopy, which revealed improvements in the microstructure and porosity of the concrete samples.

The important results from this research on high-performance, eco-friendly concrete are highlighted by the following conclusions.

1.  EMMA software can be successfully used to design high-performance concrete mixes with and without supplementary cementitious material for better particle packing that can result in more durable concrete compared with conventional concrete. However, newly developed concrete mixes using EMMA software yielded slightly lower compressive strength than that of the corresponding OC due to the use of large quantities of 10 mm aggregate in the new mixes. The ratio of strength at 7 days to strength at 28 days for new mixes modeled by EMMA ranged from 0.79 to 0.93.
2.  The fresh properties of the developed concrete with proper particle packing were found by testing the temperature, air content, and slump of the mixtures. All mixes were workable and slump values ranged from 120 to 160 mm. The temperature of the concrete was about 30 °C, and the air content ranged from 1 to 1.4%.
3.  The resistance to chloride penetration tests showed that adding GGBS and SF can lead to significant decrease in concrete permeability. The pores in the eco-friendly concrete were about twofold less frequent than those in OC. Use of supplementary materials in the mixes increased the tensile strength of the concrete, as expressed in terms of the modulus of rupture, by 5–30% compared to OC. The ACI 318 code was capable of predicting the modulus of elasticity of the optimized concrete mixes within 1–9%.
4.  SEM images show an abundance of large pores, shortage of C-S-H gel, existence of ettringites, and presence of microcracks in the OC. The microstructure was greatly improved by the addition of supplementary materials GGBS and SF, which is an indication of the expected high durability performance of the eco-friendly concrete.
5.  Eco-friendly concrete can be produced in the UAE at the same price as OC. From the long-term viewpoint, use of concrete that utilizes SF and GGBS in different structures can enhance their durability and consequently their life span, which adds to the economy. The study also showed that $CO_2$ emissions and heat of hydration were significantly reduced by replacing cement with GGBS in the concrete mixes.

**Author Contributions:** Methodology, A.K.T.; Software, A.I.; Validation, M.M.E.-E.; Formal analysis, S.W.T.; Investigation, W.J.A.; Writing—original draft, T.K.M.A. All authors have read and agreed to the published version of the manuscript.

**Funding:** This research was funded by the Open Access program and College of Engineering at the American University of Sharjah. Elkem provided the software and silica fume materials used in the investigation.

**Institutional Review Board Statement:** Not applicable.

**Informed Consent Statement:** Not applicable.

**Data Availability Statement:** All data are contained within the article.

**Acknowledgments:** The authors appreciate the help provided by Arshi Shakeel Faridi, Mohamed Ansari Abdulbhasheer and CONMIX company staff during the experimental part of the study. The opinions included in the study are those of the authors and do not reflect the views of the funding agencies.

**Conflicts of Interest:** The authors declare no conflict of interest.

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
