# Peer review of "Influence of Optimum Particle Packing on the Macro and Micro Properties of Sustainable Concrete"

_sustainability, doi:10.3390/su151914331_

Round 1

Reviewer 1 Report

In this paper, the possibility of making eco-friendly concrete from available materials in the local United Arab Emirates market was investigated. Supplementary cementitious materials, such as ground granulated blast furnace slag  and silica fume , were utilized for decreasing the cement quantity, enhancing the particle size distribution and improving packing,which is beneficial to the renewable resources of construction solid waste. However, there are some problems in this paper, as follows:

1.      There is a lack of necessary test data and results in the abstract.

2.      The writing of CO2 is not standardized, for example, lines 28 and 31.

3.      There is no uniform format of the article, such as lines 156-226.

4.      The representation of the graph is not standardized,for example, line 249.

5.      Lines 314 and 315 are the same?

6.      The clarity of the picture is not good, for example,Figure 3 a,b,c,Figure 12 and Figure 13.

7.      The quality of the literature also needs further revision.

A small amount of English checking is needed.

Author Response

Thank you for your valuable comments. We are glad to send you our response, I hope you find them useful.

Kind regards,

Adil Tamimi

Reviewer 2 Report

In this manuscript, the possibility of making eco-friendly concrete from available materials in the local United Arab Emirates (UAE) market was investigated. 130 concrete specimens in the shape of cubes, cylinders, and prims from 10 different concrete mixes were tested to determine the enhancement levels in the fresh and hard properties of new concrete. The study also demonstrated the significant decrease in CO2 emissions of concrete utilizing GGBS and the financial feasibility of eco-friendly concrete in the UAE. However, the research work is not very revealing and the results are not soundly innovative. The following comments should be considered:

 1. The Introduction section is too much and it is recommended that it be reorganized.

2. The contents of lines314 and line315 are duplicated.

3. Figure 3 is too fuzzy to see any test details, such as air content, slump-specific values, etc.

4. In Table 8, what are the units of the TRCP values? Is the specific photo of the TRCP test the RCPT of Figure 3(f)? TRCP and RCPT are easily confused and can they be harmonized?

5. Figure 7 has low pixels and local enlargement is recommended if the shear-crushing surface of the aggregate is to be represented.

6. What is the exact process of material preparation? What kind of machine is used? What is the mixing speed?

7. What is the exact value calculated for ACI 318 on lines 433-436? Similarly, in lines 448-450, for the MOE calculation, what is the specific value?

8. Figure 11 has low pixels, including the scale icons. Similarly, figures 12 and 13, both of which have very low pixels, (a) should retain the scale icons, and (b)-(e) are too small to read.

9. The conclusion is too long and should be reorganized to clearly express the main conclusions and innovations.

Author Response

Dear Reviewer,

Thank you for providing valuable comments. We are happy to respond to your comments. 

Best regards,

Adil Tamimi

Reviewer 3 Report

1. The abstract was written in a general tone that needs to be supported by more results. There is no need for sentences that give general information such as those at the beginning of the abstract.

2. Table 2 includes the chemical composition of cement and GGBS. What about silica fume?

3. Figure 1 includes the PSD of the GGBS material and various aggregates. What about cement and silica, when I add them, they are important.

4. The author/s wrote in Line 314 Table 3: Chloride permeability limits [27]. and in Line 315  Table 3: Chloride permeability limits [40]. Correct.

5. Compare the results obtained with results from previous research to show the originality of the research

6. Figures 12 and 13 b to e are not clear. Replace them with readable ones

7. Add some ratios and percentages of increase and decrease to the conclusions section, showing the originality of your conclusions.

Author Response

Dear Reviewer,

Thank you for your valuable comments. We are happy to respond to your comments.

Best regards,

Adil Tamimi

Reviewer 4 Report

The feasibility of producing eco-friendly concrete from locally available materials in the United Arab Emirates market was investigated in this study. Supplementary cementitious materials, such as ground granulated blast furnace slag and silica fume, were used to reduce cement quantity and improve particle size distribution.

Technical comments:

1- Introduction is too long. Text should be reduced; however, this is left up to the authors’ discretion.

2- In section 2, enhancement in the rheology was sought; however, the rheological properties (cement hydration and chemical interactions in the cement paste system) of the concrete were not address. A discussion should be provided.

3- The parameter q value is considered to have a certain relationship with the rheological properties of the concrete slurry, and eventually the q values affects the mix flow properties. How did the author arrive at q value of 3?

4- What is the optimal mix: the optimal ratio between the raw materials?

5- EMMA does not account for superplasticizer effectiveness; therefore, some adjustments to superplasticizer dosage or workability may be required when concrete is prepared. What adjustments did the authors make to obtain the optimal mix?

Author Response

Dear reviewer,

Thank you for your valuable comments. We are happy to provide our response to your comments.

Best regards

Adil Tamimi

Round 2

Reviewer 4 Report

The reviewers' comments are addressed satisfactorily.